# Food Environment in the Lower Mississippi Delta: Food Deserts, Food Swamps and Hot Spots

**DOI:** 10.3390/ijerph17103354

**Published:** 2020-05-12

**Authors:** Melissa Goodman, Jessica Thomson, Alicia Landry

**Affiliations:** 1U.S. Department of Agriculture, Agricultural Research Service, Stoneville, MS 38776, USA; hallgoodman@gmail.com; 2Department of Family and Consumer Sciences, University of Central Arkansas, Conway, AR 72035, USA; alandry@uca.edu

**Keywords:** food environment, spatial analysis, hot spot analysis, food swamps, food deserts, diet quality

## Abstract

The objectives for this study were to examine the location and density of measured food outlets in five rural towns in the Lower Mississippi Delta, determine the spatial location of Delta Healthy Sprouts (DHS) participants’ homes in the food environment, and examine relationships between the spatial location of participants’ homes and their diet quality. Using a food desert/food swamp framework, food outlet geographic locations were analyzed in relation to one another, the distance between DHS participants’ residence and closest food outlets by class were computed, and associations among residents’ diet quality, hot spot status, and census tract classification were explored. Of 266 food outlets identified, 11 (4%), 86 (32%), 50 (19%), and 119 (45%) were classified as grocery stores (GS), convenience stores (CS), full-service restaurants (FS), or fast food restaurants (FF), respectively. A third of participants lived in CS hot spots, while 22% lived in FF hot spots. DHS participants lived closer in miles to CS (0.4) and FF (0.5) as compared to GS (1.6) and FS (1.1) outlets. Participants bought most groceries at national chain grocery stores rather than their closest grocery store. The food environments of the five towns and associated neighborhoods in which DHS participants resided were not supportive of healthful eating, containing both food deserts and food swamps, often in overlapping patterns.

## 1. Introduction

The United States (U.S.) Department of Health and Human Services and the U.S. Department of Agriculture [1] estimate that about half of adults in the U.S. have one or more preventable, diet-related chronic diseases, including cardiovascular disease, type 2 diabetes, and overweight or obesity. In the U.S., publicly available dietary guidelines based on the most current scientific and medical knowledge are published every five years, yet many adults in the U.S. fall short of these dietary recommendations [2]. A number of approaches have been taken to improve diets in the U.S., including mandating healthy foods in school lunchrooms, educating individuals and the public about healthy eating, and improving access to healthy foods. These efforts have been met with varying degrees of success.

Due to the well-established links of diet with obesity, chronic disease, and premature death [3,4,5,6], there is great interest in finding ways to improve diets and food selection. The food environment has been identified as one factor influencing food selection and thus, diet quality. Food deserts and food swamps have emerged as key concepts in the modern food environment vocabulary. Food deserts are locations where residents lack access to healthy foods. Based on a systematic review of the food desert literature, Beaulac and associates concluded that Americans living in low-income and minority areas tend to have poor access to healthy food [7]. They further concluded that structural inequities in the food retail environment may contribute to inequalities in diet and diet-related outcomes [7]. Elements of the food desert definition include distance to the closest grocery store, poverty of geographic location, urban/rural location, and access to a vehicle. The USDA Economic Research Service (ERS) developed and supports the Food Access Research Atlas, an interactive national atlas to assist planners and researchers in describing access to supermarkets in specific geographical locations [8]. In contrast, food swamps are defined as areas with an excess of unhealthy foods (e.g., energy-dense junk and snack foods) in relation to healthy foods (e.g., fruits and vegetables) [9]. Close proximity to unhealthy food outlets has been associated with poor diet [10,11]. Elements of a food swamp include clusters of food retailers such as convenience stores and fast food restaurants. Food desert and food swamp orientations offer different policy solutions for improving diet quality. A food desert focus sees poor diets as partially being the result of low access to healthy foods. This focus is reflected in policies such as the Healthy Food Financing Initiative, which provides incentives to build grocery stores in food deserts. A food swamp focus sees plentiful access to high-calorie foods as the key to poor diets. This focus is reflected by zoning policies, such as city ordinances that limit the building of new convenience stores and fast food restaurants in identified areas. Much of the current research on food deserts and swamps has focused on policy evaluation to determine if policies have the intended effects on improving the diets of those living in these areas.

We previously conducted Delta Healthy Sprouts (DHS), a nutrition and physical activity intervention designed for pregnant women and their infants residing in the Lower Mississippi Delta [12], a region characterized by high rates of obesity, chronic disease, and premature death [13,14,15]. Because we suspected that the nutrition environment to which these women and their infants were exposed played a role in their persistently low diet quality [16,17], we collected observational data on the food environments of the towns in which these women lived. Results from the Delta Food Outlets (DFO) study indicated that nutrition environment scores were generally low for all classes of food outlets (grocery stores, convenience stores, full-service restaurants, and fast food restaurants) and many outlets provided little in the way of healthy options [18]. The objectives of the present paper were to: (1) examine the physical location and density of the measured food outlets using spatial analysis combined with a food desert/food swamp framework; (2) determine the spatial location, including proximity to food outlets, of DHS participants’ homes in the food environment; and (3) examine relationships between the spatial location of participants’ homes in the food environment and their diet quality.

## 2. Materials and Methods 

The DFO Study was an ancillary study of the DHS Project that was conducted to collect observational data on the food environment in the five towns where the DHS participants resided. Data collection for the DFO Study occurred from March 2016 to September 2018. The DFO study was exempted from review by the Delta State University Institutional Review Board (protocol number 16-001).

Food outlets included in this study were grocery stores, convenience stores, full-service restaurants, and fast food restaurants. Mobile food retailers, roadside food retailers, event vendors, specialty food retailers, and retailers that were not open on regular hours most days of the week (e.g., farmers markets) were not included. The operational definition of food outlets used for this study were adopted as those provided by USDA ERS in the documentation of the Food Environment Atlas [19]. Grocery stores are defined as “supermarkets and smaller grocery stores primarily engaged in retailing a general line of food, such as canned and frozen foods; fresh fruits and vegetables; and fresh and prepared meats, fish, and poultry”. Convenience stores “are primarily engaged in retailing a limited line of goods that generally includes milk, bread, soda, and snacks”. Full-service restaurants are characterized by “providing food services to patrons who order and are served while seated (i.e., waiter/waitress service) and pay after eating. These establishments may provide this type of food service to patrons in combination with selling alcoholic beverages, providing takeout services, or presenting live nontheatrical entertainment”. Fast food restaurants are characterized by “providing food services (except snack and nonalcoholic beverage bars) where patrons generally order or select items and pay before eating. Food and drink may be consumed on premises, taken out, or delivered to the customer’s location. Some establishments in this industry may provide these food services in combination with alcoholic beverage sales”.

Grocery stores were identified by referencing the USDA Food and Nutrition Service’s Supplemental Nutrition Assistance Program (SNAP) retailer locator that contains lists of SNAP retailers by state, as well as location [20], and the Mississippi State Department of Health’s (MSDH) Restaurant and Food Facility Inspections website [21]. Convenience stores were identified in a variety of ways. The USDA Food and Nutrition Service’s SNAP retailer locator was referenced [20]. A second source of identification was the B2B Yellow Pages website that contains a search engine for business type and city/state location [22]. Another source for identification was the current privilege license list obtained from the City Clerk in four of the towns. Each convenience store was coded as either a corner store, gas station, chain dollar store, or chain pharmacy. Restaurants were identified by referencing the MSDH online Restaurant and Food Facility Inspections website [21]. Restaurants cannot operate without a current license and passing annual health and sanitation inspection. After the completion of DFO audits, DFO staff classified restaurants as either full-service or fast food. In addition, each fast food retailer was coded as either a stand-alone fast food restaurant, a corner store (convenience store) that sells fast food, a gas station (convenience store) that sells fast food, or a grocery store that sells fast food (delicatessen).

The ERS Food Access Research Atlas 2015 food desert definition, based on income and access (distance to a grocery store), was used to classify town census tracts as: not low income and low access (>1.0 mile, NLILA), low income and not low access (≤0.5 mile, LINLA), low income and moderate access (LIMA), and low income and low access (LILA) [8]. Census tracts classified as LILA were defined as food deserts. Food swamps were defined as clusters of convenience stores and clusters of fast food restaurants. These clusters were identified using a spatial analysis technique called hot spot analysis.

An online geographic tool, Latitude/Longitude Finder [23], was used to convert physical addresses to latitude and longitude coordinates and to visually verify that the location was correctly identified on a map. Additionally, all locations were physically visited to further verify that the map location was correct and that the outlet was currently operational and selling food. Distances from the DHS participants’ residence to the closest grocery store, full-service restaurant, fast food restaurant, and convenience store were measured using the Network Analysis Closest Facility tool in ArcGIS (version 10.4.1, ESRI, Redlands, CA, USA). All distances were road distances measured in miles.

At baseline, DHS participants completed several surveys and provided 24-hour dietary recalls [12]. Data pertinent to this study included outlets where most food was purchased, car ownership, and diet quality. Outlets where the most food was purchased over the past month was ascertained with the following question: “Including yourself and anyone else in your household who shops for food, where was most of the food bought?” Participants provided the store name and address, including the town. Vehicle ownership was assessed with the following question: “Do you own or have access to a car?” The three response options were yes – own, yes – access but not own, and no. None of the towns in which the participants lived had public transportation. Dietary intake was collected via multiple-pass 24-hour food recalls using Nutrition Data System for Research (NDSR) software [24]. Based on data from the food recalls, total and component scores from the Healthy Eating Index 2010 (HEI 2010) were computed [25]. The HEI 2010 measures adherence to the 2010 Dietary Guidelines for Americans [26] and includes 12 components that are summed to create a total score from 0 to 100 points. The 12 components include total vegetables, greens and beans, total fruit, whole fruit, whole grains, dairy, total protein foods, seafood and plant protein foods, fatty acids, refined grains, sodium, and empty calories. For each component, higher scores reflect greater adherence to 2010 Dietary Guidelines for Americans. Details regarding HEI 2010 components, maximum points, standards for a maximum score, and standards for a minimum score of zero can be found elsewhere [25]. Only the 68 DHS participants who lived within one of the five towns were included in the present analyses. The remaining 14 participants were excluded because they lived outside town boundaries.

Geographical analyses were conducted using ArcGIS. We focused on density and proximity, two important dimensions of the food environment [27]. All locations (grocery stores, convenience stores, and restaurants) and participants’ home addresses were geocoded with latitude and longitude coordinates. The ArcGIS Spatial Autocorrelation tool which calculates Global Moran’s I statistic was used to determine whether patterns of establishments (i.e., convenience stores or fast food restaurants) were clustered, dispersed, or random in the geographic areas of interest. The ArcGIS Hot Spot Analysis tool calculates the Getis-Ord Gi* statistic for specified features (e.g., convenience stores, fast food restaurants) in the dataset. The resultant z-scores and *p*-values indicated where features with either high (hot spot) or low (cold spot) values clustered spatially [28,29]. The hot spot analysis focused on the five towns using a 1000 x 1000 m Universal Transverse Mercator (UTM)-15 grid. Data were visually depicted on maps generated using ArcGIS. Primary highways were included on the maps. Primary highways are defined as those highways designated in the Mississippi 2019 Official State Highway Map legend as “principal through highways” [30]. ArcGIS Network Analysis was used to calculate road miles between food outlets and participants’ home addresses.

Statistical analyses were performed using SAS^®^ (version 9.4, SAS Institute Inc., Cary, NC, USA). Because the study was exploratory in nature and data collection encompassed all food outlets in the five towns (i.e., entire population), no sample size or power analyses were conducted. Descriptive statistics, including means, standard deviations, frequencies, and percentages were computed to characterize the various food environments. Prior to analysis, distributions of HEI 2010 total and component scores were checked for approximate normality. Three scores either passed the goodness of fit tests (Cramer von Mises and Anderson Darling) (empty calories and total) or failed the tests but appeared sufficiently normal for underlying assumptions of normality to be reasonable for practical purposes of analysis (total vegetable). The significance level for differences between groups was set at *p* ≤ 0.01 due to the number of comparisons made (13 diet quality scores). To assess group differences in HEI 2010 scores by hot spot status, one-sided two-sample t tests were used for total and component scores with approximately normal distributions, and one-sided Wilcoxon signed rank tests were used for all other component scores. One-sided tests were used because scores were expected to be lower for participants living in hot spots as compared to participants not living in hot spots. Because the NLILA and LINLA census tract classes contained only three and five participants, respectively, they were excluded from the analyses assessing group differences in HEI 2010 scores. Thus two-sided two-sample t-tests and Wilcoxon signed rank tests were used to assess differences in HEI 2010 scores between the LIMA and LILA census tract classes. Two-sided tests were used because it was not clear whether HEI 2010 scores for participants living in LIMA census tracts would be higher or lower than scores for participants living in LILA census tracts.

## 3. Results

Population characteristics and food outlet frequencies of the five focus towns are presented in Table 1. Town populations were mainly African American and had poverty rates exceeding the national average [31]. The three larger towns were classified as urban clusters (2500–50,000), while the two smaller towns were classified as rural (less than 2500) [32]. The majority of food outlets were classified as fast food restaurants, followed by convenience stores, full-service restaurants, and grocery stores. The majority of convenience stores were subclassified as gas stations, followed by corner stores, national chain dollar stores, and national chain pharmacies. The majority of fast food restaurants were subclassified as stand-alone fast food restaurants, gas stations, grocery store delicatessens, and corner stores. This resulted in 32% of fast food restaurants with dual classification as both a convenience store and a fast food restaurant (24% of corner stores and 63% of gas stations) or a grocery store and a fast food restaurant (73% of grocery stores).

Results from hot spot analysis are presented in Figure 1 (convenience stores) and Figure 2 (fast food restaurants). The majority (55%) of convenience stores were located on a primary highway with some variability by convenience store subclass. Over two-thirds of gas stations, chain dollar stores, and chain pharmacies were located on a primary highway, while only 10% of corner stores were similarly located. The majority (55%) of full-service restaurants were located on a primary highway (map not shown). Similar to convenience stores, the majority (69%) of fast food outlets were also located on primary highways, with the exception of corner stores that sold fast food (20% located on primary highways). Most hot spots fell along primary highways.

Relationships between Delta Healthy Sprouts participants’ residence and food outlets and hot spots are presented in Table 2. On average, participants lived within ½ mile from convenience stores and fast food restaurants, but over 1 mile from full-service restaurants and grocery stores. All DHS participants identified (by name) a grocery store as the place where most household food was purchased. Only 19% shopped at their closest grocery store, while 18% and 63%, respectively shopped at a local grocery store or a large national chain grocery store that was farther from their home. The mean distance to the closest grocery store was less than half the mean distance to the grocery store where most household food was purchased. Approximately one-third of DHS participants lived in convenience store hot spots, while over one-fifth lived in fast food hot spots.

Figure 3 shows the food desert classification of census tracts in the five focus towns, the location of the grocery stores, and the location of convenience store and fast food hot spots. All but three of the 11 grocery stores were located in a LIMA or LILA census tract. Characteristics of the DHS participants’ residence by census tract class are presented in Table 3. LIMA and LILA census tracts had the highest percentages of residents who were African American, as well as those living below the poverty level, as compared to NLILA and LINLA census tracts. DHS participants who lived in LIMA census tracts were, on average, closer to all four types of food outlets than participants residing in the other three census tract classes. The majority and equal proportions of DHS participants lived in LIMA and LILA census tracts, and participants in these tracts were the only ones living in convenience store and fast food hot spots. 

Associations among DHS participants’ HEI 2010 scores, hot spot status and census tract classification are presented in Table 4. No significant group differences were found between participants living in and not living in convenience store hot spots nor between participants living in and not living in fast food hot spots. Further, no significant group differences were found between participants living in LIMA and LILA census tracts.

## 4. Discussion

Results from this study indicate that the majority of DHS participants’ food environments contained both food deserts and food swamps, both of which are associated with unhealthy diets. Most food swamps (convenience store and fast food hot spots) were contained in LIMA and LILA census tracts, indicating considerable overlap between food deserts and food swamps in these five rural towns. Although we defined food deserts as LILA census tracts for the present analyses, LIMA census tracts do fit a less stringent definition of food deserts (more than ½ mile vs. 1 mile from nearest grocery store) [8]. Additionally, the LIMA and LILA census tracts had the highest percentages of African American residents, as well as those below the poverty level, which is consistent with research showing higher density of convenience stores and fast food restaurants in low-income and minority neighborhoods [10,33]. Rahkovsky and Snyder [34] found that when they separated the LILA measure into low-access areas and low-income areas, low income was more strongly associated with purchases of unhealthy food than was living in an area with limited access to supermarkets. We were not able to confirm this result in terms of diet quality in the present study due to the small number of DHA participants living in LINLA census tracts. However, it is interesting that participants living in the LINLA tracts were, on average, 2 miles from the closest grocery store, indicating that on an individual level, they did not fit their census tracts’ definition of “not low access”. These findings deserve further study in rural Lower Mississippi Delta communities.

While the majority of DHS participants lived in food deserts, they also owned or had access to a vehicle. Like most individuals living in the U.S. [35,36,37], proximity to a grocery store appeared to be of less importance because participants did not buy most of their food at the grocery store closest to their residence, but instead bought most of their groceries at a supermarket or superstore farther from their home. However, as past research has shown, shopping at supermarkets is not always associated with the purchase of healthier food [38,39]. Richardson et al. [40] found that there were improvements in the economic wellbeing and health of residents living in food deserts after a full-service grocery store was built. In contrast, Allcott et al. [41] argued that building a grocery store in a food desert, or residents driving to a grocery store with healthy food, or residents moving to an area with a grocery store with healthy food, had minor effects on diet and are not solutions for improving nutrition status. They suggest that a means-tested subsidy (based on household income) for healthy groceries could increase low-income households’ healthy eating to the level of high-income households at about 15 percent of the cost of the SNAP program.

Contrary to previous findings regarding the relationship between greater access to energy-dense food and a poor diet [42,43], living in food swamps was not associated with lower diet quality in the present study. It is possible that small sample sizes, coupled with relatively large variability in diet quality scores, were at least partly responsible for the lack of significant findings. Although the differences failed to reach statistical significance, DHS participants living in food swamps scored approximately 2 points lower for diet quality respective to empty calories than DHS participants not living in food swamps. Similar results were found in a U.S. study conducted with adolescent girls in which the authors reported that girls living in food swamps consumed more snacks and desserts than girls who did not live in food swamps [11]. The present study was not purposely designed to test for diet quality differences between residents living in and not living in food swamps, but our results suggest that differences may exist. Further research with larger sample sizes is warranted.

Similar to previous findings [7], census tracts defined as food deserts, using either the less stringent ½ mile (LIMA) or more conservative 1 mile (LILA) distance, had higher proportions of African American residents and residents living below the poverty level than non-food desert tracts. Additionally, the LIMA and LILA tracts were the only tracts containing DHS participants living in food swamps. Because of these similarities, it is perhaps not surprising that significant differences in diet quality were not found between participants living in LIMA and LILA census tracts. However, while failing to reach statistical significance, notable differences were apparent in diet quality respective to empty calories (3 points higher in LILA) and sodium (3 points higher in LIMA). Thus, relatively small sample sizes and high variability may also have resulted in non-significant findings. These results suggest that when designing nutrition interventions targeting low-income neighborhoods or residents, researchers may need to consider that differences in diet quality components exist among neighborhoods. As stated by Vaughan and associates [44], effective policies aimed at improving diets should focus on both the food environment as well as the characteristics of individuals, because both are important predictors of diet.

## 5. Conclusions

The food environments of the five towns and associated neighborhoods in which DHS participants resided contained both food deserts and food swamps, often in overlapping patterns. Despite living, on average, less than 2 miles away from a grocery store, most participants chose to shop at large chain grocery stores more than twice that distance from their homes. The lack of significant associations between diet quality and residence in food deserts and food swamps deserves further study with larger sample sizes and a purposeful design. The current study quantified the food environment of five towns in the rural Lower Mississippi Delta in terms of food deserts and food swamps. However, future studies are needed to longitudinally measure food environments and residents’ shopping habits, health behaviors, and health outcomes to better understand the socioeconomic and environmental risks and their interplay on health behaviors, such as dietary intake.

## Figures and Tables

**Figure 1 ijerph-17-03354-f001:**
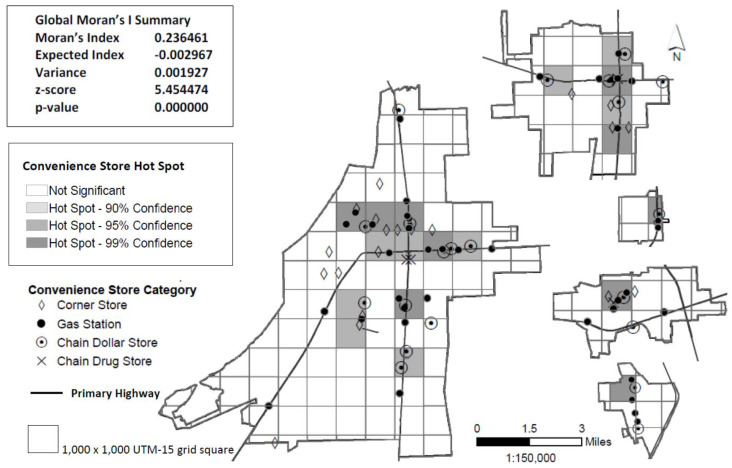
Spatial distribution and clustering (hot spots) of convenience stores in five rural Lower Mississippi Delta towns.

**Figure 2 ijerph-17-03354-f002:**
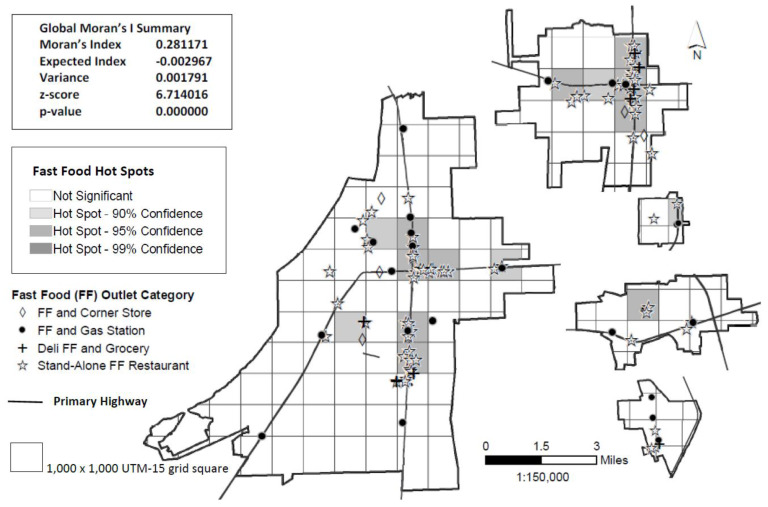
Spatial distribution and clustering (hot spots) of fast food restaurants in five rural Lower Mississippi Delta towns.

**Figure 3 ijerph-17-03354-f003:**
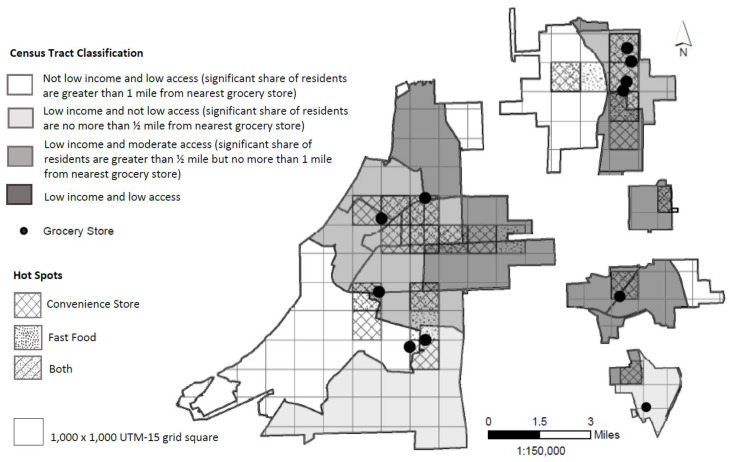
Spatial distribution of grocery stores in relation to census tract income and food access classification and hot spot type in the five rural Lower Mississippi Delta towns.

**Table 1 ijerph-17-03354-t001:** Population characteristics and food outlet frequencies of the Lower Mississippi Delta towns (n = 5).

Characteristic ^1^	United States		Town 1		Town 2		Town 3		Town 4		Town 5	
**Population (n)**			32,612		12,346		4254		2484		1750	
Black (%)	13.8		80.0		49.4		68.9		86.0		90.7	
Below poverty level (%)	15.1		36.0		28.5		32.1		41.4		50.6	
	**Overall**		**Town 1**		**Town 2**		**Town 3**		**Town 4**		**Town 5**	
**Food outlet class**	**n**	**%**	**n**	**%**	**n**	**%**	**n**	**%**	**n**	**%**	**n**	**%**
Grocery store	11	4.1	5	45.5	4	36.4	1	9.1	1	9.1	0	0.0
Convenience store	86	32.3	48	55.8	18	20.9	10	11.6	6	7.0	4	4.7
Full-service restaurant	50	18.8	28	56.0	16	32.0	5	10.0	1	2.0	0	0.0
Fast food restaurant	119	44.7	60	50.4	40	33.6	8	6.7	7	5.9	4	3.4
Convenience store subclass												
Corner store	21	24.4										
Gas station	40	46.5										
Chain dollar store	21	24.4										
Chain pharmacy	4	4.7										
Fast food restaurant subclass												
Corner store	5	4.2										
Gas station	25	21.0										
Grocery store delicatessen	8	6.7										
Restaurant stand-alone	81	68.1										

^1^ Based on 2012–2016 American Community Survey 5-year population estimates [28].

**Table 2 ijerph-17-03354-t002:** Distances between Delta Healthy Sprouts participants’ residence and food outlets and frequencies of residences in food outlet hot spots.

	Distance between Residence and Food Outlet					
	Miles			% Living Within		
Food Outlet Class	Range	Mean	SD	1/4 Mile	1/2 Mile	1 Mile
Grocery store						
Closest	0.4–11.1	1.6	2.13	0.0	5.9	44.1
Most food purchased ^1^	0.4–22.9	4.2	5.15	0.0	1.6	14.5
Convenience store	<0.1–1.1	0.4	0.23	29.4	76.5	98.5
Full-service restaurant	0.1–10.0	1.1	1.97	11.8	30.9	79.4
Fast food restaurant	0.1–1.5	0.5	0.25	19.1	66.2	95.6
	**Frequency of residences within hot spot**					
	**In**		**Adjacent ^2^**		**Not in/adjacent**	
**Hot spot type**	n	%	n	%	n	%
Convenience store	22	32.4	29	42.6	17	25.0
Fast food restaurant	15	22.1	22	32.3	31	45.6

SD, standard deviation. ^1^ 6 participants did not provide information.^2^ Adjacent cell shares a complete side border with hot spot cell.

**Table 3 ijerph-17-03354-t003:** Characteristics of Delta Healthy Sprouts participants’ residence by census tract class.

	Census Tract Class							
	NLILA		LINLA		LIMA		LILA	
Census Tract	Mean	SD	Mean	SD	Mean	SD	Mean	SD
African American (%)	34.0	15.56	64.1	6.64	91.3	12.01	82.3	10.38
Below poverty level (%)	13.3	0.05	26.3	7.00	51.3	12.84	36.4	8.05
**Participant Residence**								
Distance (miles) to closest								
Grocery store	1.3	0.27	2.0	1.18	0.9	0.34	2.2	3.00
Convenience store	0.5	0.28	0.6	0.25	0.3	0.13	0.5	0.24
Full-service restaurant	0.7	0.27	1.1	0.72	0.6	0.30	1.7	2.80
Fast food restaurant	0.6	0.31	0.8	0.42	0.3	0.13	0.5	0.22
	**n**	**%**	**n**	**%**	**n**	**%**	**n**	**%**
Overall	3	4.4	5	7.4	30	44.1	30	44.1
Convenience store hot spot	0	0.0	0	0.0	13	43.3	9	30.0
Fast food restaurant hot spot	0	0.0	0	0.0	9	30.0	6	20.0

NLILA, not low income and low access; LINLA, low income and not low access; LIMA, low income and moderate access; LILA, low income and low access; SD, standard deviation.

**Table 4 ijerph-17-03354-t004:** Healthy Eating Index 2010 component and total scores by hot spot status and census tract classification.

	Convenience Store Hot Spot				
Component	No (n = 46)		Yes (n = 22)		
(score range)	Mean	SD	Mean	SD	*p* ^1^
Total vegetable (0–5)	1.9	1.37	2.2	1.44	0.790
Empty calories (0–20)	11.8	5.78	9.2	5.44	0.044
Total (0–100)	42.8	14.10	42.8	9.75	0.500
	**Median**	**IQR**	**Median**	**IQR**	***p*^2^**
Greens and beans (0–5)	0.0	0.00	0.0	0.00	0.487
Total fruit (0–5)	0.0	2.49	0.1	3.01	0.254
Whole fruit (0–5)	0.0	2.47	0.0	4.67	0.273
Whole grains (0–10)	0.0	3.40	0.0	4.27	0.402
Dairy (0–10)	3.4	6.04	3.5	2.90	0.435
Total protein foods (0–5)	5.0	0.78	5.0	1.23	0.357
S&P protein foods (0–5)	0.0	0.89	0.0	0.00	0.023
Fatty acids (0–10)	6.3	7.32	5.0	7.21	0.310
Sodium (0–10)	1.8	4.79	2.9	5.21	0.155
Refined grains (0–10)	5.8	5.36	6.9	4.82	0.097
	**Fast food hot spot**				
	**No (n = 53)**		**Yes (n = 15)**		
	**Mean**	**SD**	**Mean**	**SD**	***p*^1^**
Total vegetable (0–5)	1.9	1.39	2.3	1.41	0.839
Empty calories (0–20)	11.3	6.13	9.5	4.01	0.141
Total (0–100)	42.9	13.63	42.4	9.57	0.447
	**Median**	**IQR**	**Median**	**IQR**	***p*^2^**
Greens and beans (0–5)	0.0	0.00	0.0	0.00	0.205
Total fruit (0–5)	0.0	2.30	2.3	5.00	0.059
Whole fruit (0–5)	0.0	1.90	0.0	4.94	0.181
Whole grains (0–10)	0.0	4.46	0.0	4.02	0.465
Dairy (0–10)	3.3	5.55	3.7	2.64	0.500
Total protein foods (0–5)	5.0	0.78	5.0	4.61	0.278
S&P protein foods (0–5)	0.0	0.89	0.0	0.00	0.015
Fatty acids (0–10)	6.4	7.24	4.5	8.28	0.137
Sodium (0–10)	2.1	4.79	1.9	5.21	0.388
Refined grains (0–10)	6.1	6.58	6.9	4.34	0.196
	**Census tract**				
	**LIMA (n = 30)**		**LILA (n = 30)**		
**Component**	**Mean**	**SD**	**Mean**	**SD**	***p*^3^**
Total vegetable (0–5)	1.7	1.25	2.3	1.54	0.132
Empty calories (0–20)	9.6	6.46	12.8	4.57	0.028
Total (0–100)	41.3	13.57	45.0	10.98	0.243
	**Median**	**IQR**	**Median**	**IQR**	***p*^4^**
Greens and beans (0–5)	0.0	0.00	0.0	0.00	0.278
Total fruit (0–5)	0.0	2.30	0.1	4.40	0.072
Whole fruit (0–5)	0.0	0.00	0.0	3.03	0.029
Whole grains (0–10)	1.3	5.09	0.0	3.67	0.356
Dairy (0–10)	2.9	4.35	3.7	4.96	0.830
Total protein foods (0–5)	5.0	1.61	5.0	0.78	0.583
S&P protein foods (0–5)	0.0	0.00	0.0	0.89	0.243
Fatty acids (0–10)	5.7	7.90	6.5	6.38	0.396
Sodium (0–10)	3.5	4.10	0.0	3.62	0.015
Refined grains (0–10)	7.0	4.91	5.7	6.74	0.598

SD, standard deviation; IQR, interquartile range (difference between 25th and 75th percentiles); S&P, seafood and plant; LIMA, low income and moderate access; LILA, low income and low access (food desert). ^1^
*p*-value for one-sided two-sample t test. ^2^
*p*-value for one-sided Wilcoxon rank-sum test. ^3^
*p*-value for two-sided two-sample t test. ^4^
*p*-value for two-sided Wilcoxon rank-sum test.

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
