# Peer review of "Food Environment in the Lower Mississippi Delta: Food Deserts, Food Swamps and Hot Spots"

_ijerph, 2020, doi:10.3390/ijerph17103354_

Round 1

Reviewer 1 Report

The aims of this study are two-fold: a descriptive component to describe the food outlets spatial characteristics in the study area, and an exploratory component to evaluate the association between food outlets spatial characteristics and residents' self-reported diet quality. Overall, I believe this is an important and well-conducted study. The methodology was solid and appropriately described. The results were clearly presented. The manuscript was clearly written. I especially appreciate that the authors used multiple sources to identify different types of food outlets to guarantee the comprehensiveness and accuracy of the identification. I also appreciate the authors’ analyses on the association between the food desert classification defined by ERS Food Access Research Atlas and the hot spots defined by the authors (Table 2), because ERS data are widely acknowledged and used nation-wide. I only have one minor comment: the authors might need to use a stricter significance threshold (P-value) in their Table 4 – they are now running into ‘multiple comparisons/testing problem’ because they are conducting individual tests on a large number of diet component measures as outcomes simultaneously which increased their risk of having Type I error.

Reviewer 2 Report

This manuscript is a fairly well-written description of a potentially very useful piece of research. The study that was conducted to assess access to food and access to quality food in the Delta region seems well designed and suitably conducted. But the add on study that is described here is not so clearly presented. There are a number of ways that the manuscript can be strengthened and be made more effective.

First, it would be helpful to have a statement that clearly states the research question and/or hypothesis/es. There doesn't seem to be a review of previous research on food desert/food swamp research upon which to base the value of the research question.  Right now, this manuscript describes a study that was done to extend a previous study, but there is no way to gauge whether the study has produced any new information or whether it gives us more understanding than we already had of deserts/swamps.

There seems to be a lot of focus on nutrition (as implied by the sentence in the introduction: "We used this information to determine proximity of DHS participants’ homes to food outlets and to examine relationships between spatial location of participants’ homes in the food environment and their diet quality." Shouldn't you also check the quality of foods that are available in each of the establishments? ... and I don't suggest you do, but clearly shopping is a choice-making task and you can buy junk food at supermarkets just as you can buy apples at convenience stores. So, the question of access to good food is not the same as acquisition of good food. There is also the question of prices of foods (which may explain why some choose to shop farther from home despite intervening opportunities. So, is the question just about availability of healthy options?  Or is it about the reality of access to healthy options?

The "Materials/Methods" section is a bit chaotic and makes for difficult reading. What I see is that you have three processes: identifying, mapping, and evaluating the distribution of food-related businesses, surveying the sample for food-buying activities and dietary information, and locating respondents' homes and analyzing their relationships to the businesses where they might get food. It seems to me that if you organized your methods more clearly on these three processes, it would seem so intertwined. 

Most of the first paragraph of this section is about the previous study, not about this study.  It should all be in the introduction, in my opinion.

It might be helpful to provide tables of the information instead of the detail of every term.  You could list the terms like "food outlet," "grocery store," "convenience store," etc. and provide the textual definition and things included in the definition, instead of writing this all up in the text.  The 2nd and 3rd paragraphs could stress the method of identifying them and geo-locating them for the GIS.

In the fourth paragraph, you provide a definition of food desert(?) as "not low income (NLI), low income (LI; low income and between 0.5 and 1.0 mile from a grocery store), and low income/low access (LILA; low income and greater than 1.0 mile from a grocery store)." If NLI is a tract that is not low income, but a tract that is LI, but within 1/2 mile of a grocery store, then what is that? It's not low income?  But it is? I "NLI" a food desert too? But no distance is given? Very confusing.

The sixth paragraph (of the Materials/Methods section) seems to be about the previous study (the DHS). It seems that this is background and should be in the introduction or a background section ... whichever works.  But, you had asked whether participants own automobiles or had access to automobiles.  What about public transportation? Are there bus systems in any of these towns? (I am guessing that there aren't).

The final paragraph in this section is about the statistical analyses.  I don't see any problems there.

The Results section should be tighter -- The initial sentence in each paragraph says that there's a table or a figure coming and "there's some stuff in it." And then you proceed to restate the contents of each of those things. There is no need for these sentences. You should make substantive statements about the results that you glean from the data.  The tables and figures are supposed to provide the details for the meaningful things that you are saying in the text. The tables and figures can be simply noted (as a parenthetical "(Table 1)" which says this is where you'll find the evidence to support my statement). The text should emphasize the most important observations, not reiterate the numbers. If you're just telling us what the percentages are from the table, why have the table?

Just a side question: Why are the four towns not identified?  Is this standard practice? Surely it can't be a matter of protecting "anonymity" of the municipalities... The Delta's a relatively small region and the maps and populations can be used to identify the communities.  It would simply be easier to mentally hold onto the patterns if we have more specific names than "Town 1, Town 2, Town 3, and Town 4."

Instead of organizing the results around each of your "exhibits," I think the results should report the answers to the questions you're "asking" (which you haven't done yet, but I'm imagining you had): What is the distribution of the food businesses and are they clustered?; Where do people shop? and Where are the places they shop in relation to their homes?; [(I guess) Where are the food deserts and swamps?]; and How do the dietary outcomes relate to the locations of shopping for foods?

The Discussion -- This section should also be organized around the major questions (same order as in the Results and hopefully way back in the discussion and literature review ... if it's added) and the discussion should stress what you've learned from the results. The current discussion is sort of weaved between paragraphs, not clear where the new set of thoughts begins or ends.

The Conclusion should reiterate the questions, the answers, the results, their meaning, and perhaps state what is still unclear or unknown.

Reviewer 3 Report

The manuscript “Food environment in the Lower Mississippi Delta: 2 food deserts, food swamps and hot spots” measures food environments of five rural town in lower Mississippi delta to study the associations between the spatial location and diet quality of Delta Healthy Sprouts participants. This manuscript pointed out that to better improve the diet quality, it is important to consider the food environment as well as individual’s characteristic. The manuscript is well written. Here are my specific comments.

  1. In the method part, the author did not provide details about participants. What is the size of DHS participants and do all DHS participants attended the survey? What is the distribution of age and poverty level? Are they the representative of the chosen population?
  2. In the manuscript, the author discussed mostly the location of the participant house as well as the distance between the house and food outlet. It is natural to think most people visit the nearby store, but since the study collect information on where the participant visited most for food. Could the author also give a short discussion on that. For example, the author can talk about the distribution of the distance between the most visited store and the residence.
  3. The author could also talk about the relationship between total scores and the classification of most visited store. This could provide more information regarding individual habits.

Round 2

Reviewer 2 Report

I have no more comments, beyond those that they chose not to accommodate.